# Morphological and Behavioral Adaptations of Silk-Lovers (Plokiophilidae: *Embiophila*) for Their Lifestyle in the Silk Domiciles of Webspinners (Embioptera)



**Thies H. Büscher** [1,*]**, J. René Harper** [2]**, Neeraja Sripada** [2]**, Stanislav N. Gorb** [1]**, Janice S. Edgerly** [2] **and Sebastian Büsse** [1]

1    Department of Functional Morphology and Biomechanics, Zoological Institute, Kiel University, 24118 Kiel, Germany
2    Department of Biology, Santa Clara University, Santa Clara, CA 95053, USA
*    Correspondence: tbuescher@zoologie.uni-kiel.de

**Abstract:** The diversity of true bugs gave rise to various lifestyles, including gaining advantage from other organisms. Plokiophilidae are cimicomorphan bugs that live in the silk constructions of other arthropods. One group, *Embiophila*, exclusively settles in the silk colonies of webspinners (Embioptera). We investigated the lifestyle of *Embiophila* using microscopy to study the micromorphology and material composition of the leg cuticle, choice assays and retention time measurements based on different characteristics of the embiopteran galleries and tilting experiments with different substrates to quantify the attachment performance of the bugs. *Embiophila* neither explicitly preferred embiopteran presence, nor required silk for locomotion, but the bugs preferred fibrous substrates during the choice experiments. The hairy attachment pad on the tibia showed the best attachment performance on substrates, with an asperity size of 1 µm. Additionally, very rough substrates enabled strong attachment, likely due to the use of claws. Our findings suggest that *Embiophila* settle in galleries of webspinners to benefit from the shelter against weather and predators and to feed on mites and other intruders. The combination of behavioral and functional morphological experiments enables insights into the life history of these silk-associated bugs, which would be highly challenging in the field due to the minute size and specialized lifestyle of *Embiophila*.

**Keywords:** cuticle composition; ecomorphology; Hemiptera; tarsus; adhesive structures; insect behavior; attachment systems

## 1. Introduction

Many animals seek shelter in different kinds of domiciles. Whereas the majority of animals use temporary constructions for protecting themselves against environmental influences, there are also several examples of construction of permanent shelters [1]. However, only a few animals produce the material for their homes by themselves; examples include primarily invertebrates building these shelters by their own secretions e.g., butterflies [2–4] and crickets [5]. Such domiciles can be useful for colony building in social insects, such as bees and wasps [6], but also for solitary predators to catch prey, as in spiders [7]. Webspinners (Embioptera) represent one peculiar case of colony-building insects [8–10]. These insects are known for building their domiciles (=galleries) using silk secretion from their own fore tarsi [11–14], in which they forage and reproduce [8,9,15]. Such galleries aid in establishing a secured colony from changing environmental conditions and especially predators for webspinners [9,16]. The said benefits of embiopteran silk galleries attract other species that make use of the sheltered hideouts. For the comparably poorly studied group of webspinners with their highly specialized domiciles, there are just few examples known of other animals specifically entering the galleries and living in the galleries with the webspinners. Just a few parasites and parasitoids (e.g., bristle flies, [17];

braconid wasps, [18]; and sclerogibbid wasps, [19,20]) are known to be associated with Embioptera. However, there is one enigmatic group of insects with a strong association to webspinners as a host: the silk-lovers from the family Plokiophilidae. Silk-lovers are true bugs (Hemiptera: Heteroptera), which all use sucking-piercing mouthparts [21]. Plokiophilidae are nested within Cimicomorpha [22–28]; they are diverse and have various feeding preferences that include extraction of plant sap, preying on other small invertebrates or hematophagy [23,29]. Plokiophilidae, in particular, are generally very host-specific. They are called silk-lovers, or web-lovers, as they inhabit the webs of spiders or webspinners, with just one known free-living exception [30]. The group currently consists of nine genera and 20 species, which are all small, ranging from 1.2 to 3.0 mm in length [31]. However, despite their largely different hosts, Plokiophilidae are considered monophyletic [25] and their members are strictly separated by their preferred hosts [32]. Most Plokiophilidae inhabit the webs of spiders (Heissophilinae and most Plokiophilinae) of the groups Mygalomorphae [33,34] and Araneomorphae [35,36]. One group within Plokiophilinae, in contrast, inhabits the silken domiciles of webspinners [32]: *Embiophila* China, 1953. This genus belongs to Embiophilina, which includes only another genus, *Paraplokiophiloides* Schuh, Štys and Cassis 2015. The latter, however, does not show this behavior.

Attachment, in general, is of importance for the locomotion of animals. To securely locomote on different substrates animals employ different kinds of adhesive systems and the functionality of any of these is constrained by the properties of the substrate [37]. Unrelated to the animal lineage of concern, including invertebrates [38] and vertebrates [39], the design of the attachment apparatus is consequently adapted to the specific substrates they encounter in their respective environments [37]. For Embiophilina the most dominant substrate is the silk of Embioptera, whose galleries they inhabit. In contrast to silk of prey capturing invertebrates, embiopteran silk is not intended to capture insects and, hence, is not covered with sticky substances, but is arranged in fine fiber bundles that form dense walls in the galleries [40,41] and is capable of transformation into thin films after contact with water [42]. As Embiophilina exclusively settle the silk galleries of webspinners, they are probably specifically adapted to this special kind of substrate. The feet of insects, in general, show various attachment devices adapted to different surfaces [43–45], including claws for interlocking with rough surfaces and adhesive pads aiding in attachment to a spectrum of finer roughness and other surface characteristics [37]. This includes different solutions to cope with the very different dimensions and challenges of the respective habitats to which the taxa are adapted [37]. Often the tarsal morphology is tuned to a specific problem or to facilitate certain tasks in which anchorage is required [46]. Especially, animals that are specialized to inhabit certain challenging habitats or substrates, such as the leaves of water-lilies [47], leaves covered with crystalline waxes [48], or the complex surfaces of the host in case of parasites [49–52], usually possess strongly specialized attachment systems. In case of such predictable substrate characteristics as a result of specialization to a defined habitat, the tarsal morphology often includes devices that reflect the lifestyle of the animal. This would be expected to be the case in the specialized silk-lovers as well.

Embiophile plokiophilids live with their silk-spinning hosts in their silken domiciles and are considered inquilines. Based on observations of four species of plokiophilids associated with embiopteran silk, Carayon [35] found the bugs to be fairly generalized in their feeding habits. Those living within webspinner silk were seen to rarely feed on the host's eggs and to more likely feed on mites and recently dead webspinners. Carayon [35] noted that bugs living in spider silk often feed on prey killed by their hosts. The inquilines with embiopterans have been seen feeding on early-instar nymphs [53], but this appears to be rare compared to cases where they feed on webspinners deceased for other reasons. These observations align with reports by Callan [54] and Ross [8]. In any case, it has been not clear whether the webspinner nymphs used as food had died of other causes. Ford [22] studied *Embiophila myersi* China, 1953 in the laboratory and additionally noted that the bugs feed on mites and sometimes are cannibalistic. She suggested that because the bugs are so small relative to their webspinner hosts, predation is very unlikely. Indeed, she

observed that the bugs did not attack their living hosts. The reliance on silk as a habitat appears to be the more persuasive argument for why the bugs live with their silk spinning hosts. Embiopteran silk might provide the same protection against the elements to the bugs as it does to the webspinners. Furthermore, Carayon [35] reported that plokiophilids can be raised for several generations without hosts as long as a fibrous substrate and mites as food are present. He reported that for three species, they can be raised in moistened fibrous cotton as well as in their host's silk. Both Carayon [35] and Ross [8] concluded that availability of silk-like substrate and food is more important than the presence of the host embiopterans as a food source ([8] see Figure 38 showing a plokiophilid sucking fluids from tarsi of a dead embiopteran).

As the tibiae of Embiophilina are all equipped with spines that resemble those of raptorial appendages [32], their actual food preferences remain in question. We investigated the life history of these enigmatic bugs and particularly focused on their substrate dependence. We aimed to study the functionality of their tarsal attachment system and their searching behavior for typical embiopteran-associated gallery features to investigate whether Embiophilina seek silk, mere hiding places, or orient more toward the presence of their hosts (e.g., recent fecal pellets in silk).

The purpose of this investigation is three-fold: First, we aimed at evaluating the bugs' responses, in the short term, when given a choice of different substrate materials, such as silk, fibers or other hiding places. If silk is critical to the bugs, we expected them to locate the silk and settle. If finding a hiding place is critical, they should accept any of the substrates offered in short term trials. Second, we quantified their ability to locomote on substrates of various roughness to gauge the degree to which the tarsi of these silk walkers are specialized. Third, we investigated their tarsal morphology in order to evaluate the morphological equipment used for both behavior and locomotion ability due to the tarsus–surface interactions with the substrates.

## 2. Materials and Methods

### 2.1. Study Subjects and Laboratory Cultures

*Embiophila myersi*, our focal plokiophilid species (Figure 1A), is common in the field in Trinidad, where they are associated with colonies of *Antipaluria urichi* (Saussure, 1896) (Clothodidae) and *Pararhagadochir trinitatis* (Saussure, 1896) (Scelembiidae). Laboratory cultures of the two webspinner species have been maintained for many years at Santa Clara University (Santa Clara, CA, USA). The bugs were also present in the culture boxes, having been inadvertently collected in the field at the same time the webspinners were gathered. Lighting in the environmental chamber was 12 h light, 12 h dark to emulate neotropical day lengths. Temperature approximated 27 °C at all times. Culture boxes were moistened twice per week, at the time the webspinners were fed on Romaine lettuce and lichens harvested from *Quercus agrifolia* bark in California.

### 2.2. Choice Tests

To determine the preferences of insects, indicating the need to find silk or the need to find a hiding place, different substrate choices were offered in 30 min trials. Each test was conducted in 3D-printed arenas (40 mm × 40 mm frame with 10mm high walls) placed in a glass Petri dish, put flat on a table, with a bottom liner of clean filter paper (Figure 1B). The arena consisted of a circular center region and four quadrants separated by 10 mm high walls. Individual bugs were allowed to wander between the four quadrants: two zones with substrates and two empty zones. Ethovision XT Software (Noldus, Wageningen, The Netherlands) was employed to record and summarize behavioral choices and wandering behaviors by the bugs. To start a trial, the bug was gently placed in the central region with gateways leading to the four zones. Choices presented were: Trial 1 (*n* = 10) offered the bug a patch of clean silk, one of silk with fecal pellets (to represent recent presence of webspinners), and two empty zones. Trial 2 (*n* = 10) offered a patch of clean silk and one of fibers pulled from a cosmetic style cotton wool ball, and two empty zones. This

arrangement was designed to test Carayon's [35] idea that any fibers are acceptable, even if not composed of silk. He also used cotton wool fibers in his experiments. For Trials 1 and 2, the substrates were alternated between the top left and bottom right quadrants (as seen in the video recording) to control for possible lighting or other unexpected biases. Trial 3 (*n* = 10) included lichens in two quadrants (typical food for the Trinidad webspinners in the field) emulating natural hiding places for the bugs, and two empty zones. This last trial tested whether they wander more if silk is not discovered. Because the bugs are extraordinarily small, the Ethovision XT software had difficulty tracking them. Therefore, for Trial 3, we switched to recording their wandering and analyzing videos manually. For Trials 1 and 2, time spent in the different quadrants was recorded by Ethovision XT. Heat maps showing where they spent time were produced by the software and time along the edge (evidence of thigmotaxis) was summarized automatically by the tracking software (Figure 2). For all trials, we additionally recorded where each bug settled and whether the bug left to wander again or transitioned from one zone to another. Each individual bug was used only once.

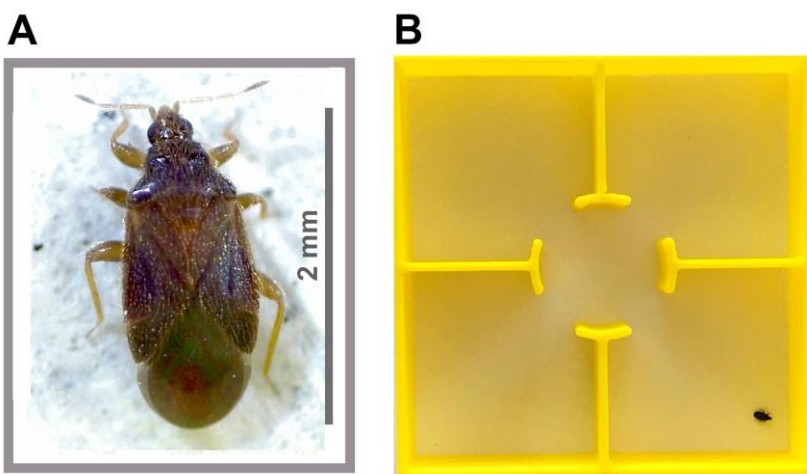

**Figure 1.** *Embiophila myersi* adult and arena for testing. (**A**) Imago, dorsal view. (**B**) 3D printed testing arena including one individual.

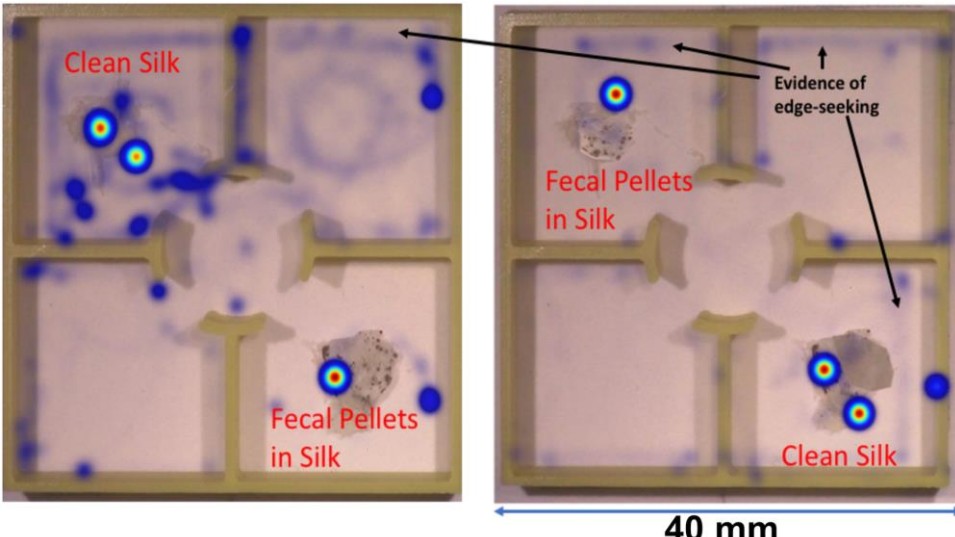

**Figure 2.** Heatmaps showing an overlay of three separate trials for each set-up (*n* = 6). Yellow and red (hot colors) indicate longer periods of time, whilst blue indicates less time in a zone. Brightly colored circles in each arena reflect the tendency for each of the bugs to settle under the silk patch once it was found. The lighter colored blue lines show, where bugs tended to walk. The blue circles show where they sat for longer periods of time.

For Trial 1 and for Trial 2, total time spent in the different zones of the arena was compared. Proportion of time spent along the edge in the different zones was similarly compared. We also determined the amount of wandering for all three set ups. Number of times the bug went into and out of the zones with substrate (lichens or the fiber samples) or without (empty) were counted and compared. After using the Shapiro–Wilk Test that revealed lack of normality in the data, we used the Kruskal–Wallis ANOVA on ranks with Wilcoxon post hoc tests adjusted for multiple comparisons with the Benjamini–Hochberg method (Section 3.3), or Wilcoxon rank sum tests corrected for multiple comparisons with the Benjamini–Hochberg method for comparisons between the time spent at the edge in the arena zones (Section 3.3) and the transitions between the zones (Section 3.4) respectively with JMP Pro 16 statistical software (SAS Institute, Cary, NC, USA) to make comparisons at a criterion for significance of $p < 0.05$.

### 2.3. Testing Attachment Performance

To determine the bug's ability to attach to substrates of various roughnesses, adult individuals ($n = 7$) were tested individually on six different substrates. Standardized industrial polishing papers with aluminum oxide coating of different grit sizes (0.3, 1.0, 3.0 and 12 µm; FibrMet discs, Buehler), sandpaper with silicon carbide coating (35 µm; Wetordry, Imperial) and clean glass were used as substrates. As the larger surface asperities of the sandpaper enabled easy engagement of the tarsi, it was used for a preceding test to confirm that each of the bugs was capable of walking easily. Presentation of the different substrates was randomized so that each bug experienced a different sequence. Glass was offered last. Once the bug was placed on the substrate and began to walk, the apparatus (a plate affixed to a rotating wheel) was tilted slowly by hand and the angle at which the bug lost its adhesion was recorded. After falling, the bug was placed in a holding container, the next substrate was set in place and the bug was tested again until all substrates were tested for that bug. The angle at which the bug started sliding on the substrate was considered the critical tilting angle. Critical tilting angles were tested for normality (Shapiro-Wilk) and equal variance (Levene) and compared using a one-way ANOVA in SigmaPlot 12.0 (Systat Software GmbH), as the data was parametric and showed homoscedasticity.

### 2.4. Scanning Electron Microscopy (SEM)

Nymphs and adults of *E. myersi* were stored in 70–80% ethanol. The samples were dehydrated in an ascending ethanol series and critical point dried using a Leica EM CPD300 (Leica, Wetzlar, Germany) CPD system. For SEM, the dried samples were sputter-coated with gold–palladium (10 nm thickness; Leica Bal-TEC SCD500, Leica Camera AG, Wetzlar, Germany), mounted on a rotatable sample holder [55] and examined in a Hitachi TM3000 (Hitachi Ltd. Corporation, Tokyo, Japan) scanning electron microscope (at 15 kV). Distances (i.e., length of setae and spatulae, width of spatulae) and circular diameters (i.e., claw tip sharpness) were measured from the micrographs, using ImageJ (v2.0.0-rc-43/1.50e, Wayne Rasband, National Institutes of Health, Bethesda, MD, USA) by using the oval tool and measuring the diameter of a circle fit into the tip of the claw ($n = 4$) and the straight line tool for linear measurements ($n = 5$ each).

### 2.5. Confocal Laser Scanning Microscopy (CLSM)

Legs of *E. myersi* (nymphs and adults) were dissected and embedded in glycerine (99.9%) on a microscope slide and covered with a high-precision cover slip. For CLSM analysis, a Zeiss LSM 700 (Carl Zeiss AG, Jena, Germany) equipped with four laser lines (wavelengths of 405, 488, 555 and 639 nm) and four emission filters (BP420-480, LP490, LP560, LP640 nm) were used (cf. Figure 1 in [56]). Maximum intensity projections were done using the ZEN2008 software (Carl Zeiss Microscopy AG, Jena, Germany). For more information on how to use CLSM to determine the material properties of the insect cuticle, we refer to Michels and Gorb [57]. Generally, CLSM analysis results in visualizations of a combined autofluorescence signal (maximum intensity projections) in every single

pixel, providing information about the presence of various components within the cuticle with different material properties [57–59]. This allows for qualitative estimates of the material composition of the analyzed cuticle. The following color code can be assigned after [57]: (1) red signal—autofluorescence of stiff sclerotized cuticle; (2) blue, green and red combined (resulting in pink, brownish, yellow and green signals within the overlay)—autofluorescence of stiff but more flexible cuticle, often chitin-dominated; (3) characteristic blue signal—autofluorescence of rubber-like cuticle with likely high proportion of resilin. The analysis herein will allow for a qualitative description only and does not represent a quantitative measurement.

## 3. Results

### 3.1. Tarsal Morphology

The studied nymphs and adults of *E. myersi* both show some differences between the leg pairs (pro-, meso- and metathoracic legs) as well as between nymphs and adults (Figures 3 and 4). In general, the legs represent regular walking legs. The tarsi are two segmented with a pretarsus and an asymmetric pair of claws; here, the medial claw is longer than the other (cf. Figure 3C). However, the tip sharpness is uniform in both claws: $679.44 \pm 67.08$ nm (mean $\pm$ SD, $n = 4$; measured for the adults). All legs are covered with uniform setae. The density of these setae increases towards the tarsus, with tibia and tarsus showing a dense coverage (Figures 3 and 4). In the maximum intensity projections obtained from CLSM, the cuticular composition seems comparably similar along the length of the respective leg, but might differ slightly between the legs of nymph and adult (Figure 5). The mentioned setae, distributed on all parts of the legs, show a green signal in both the nymphs and adults, indicating a chitinous cuticle. On both the pro- and mesothoracic legs in both nymphs and adults, strong spines are developed on the ventral surface of the femur (Figures 3 and 4). On the femur of the prothoracic leg, two rows of spines are developed in a pair with slight offset (Figure 3D). Here, three spines are over-proportionally well-developed: two in the posteroventral row of spines and one, in between, in the anteroventral row of spines (Figure 3D). On the mesothoracic legs, only one row of spines, almost in the midline of the ventrofemoral surface, is developed. These spines also differ in size: increasing towards mid-femur and decreasing again towards the femoral–tibial joint; showing two considerably large spines in the middle (Figure 4C). These spines on all legs show a brownish-red signal in the maximum intensity projections indicating a strongly sclerotized cuticle (Figure 5). Both nymphs and adults carry a row of setae facing anterior on the tibia of the prothoracic legs (Figures 3B, 4C and 5E). Almost identical structures are described in Pentatomidae as grooming devices and shown to be used in cleaning [60]. At the apical tip of the tibia, a compact row of rod-like setae is present ventrolaterally in both nymphs and adults (Figures 3B and 4B). The green signal of these rod-like setae is indicating a chitinous cuticle. The most striking difference between nymph and adult is the presence of a fibrillar adhesive pad—the tibial adhesive pad (TAP)—on the ventral surface at the apical end of the pro- and mesothoracic tibia in adults (Figure 4B,D). The TAP consists of densely packed spatulated setae (Figure 4B), showing a material gradient from reddish signal at the base over greenish towards light blue signal at the tip (Figure 5C) and an increasing length from posterior to anterior. The spatulated setae have a length of $17.39 \pm 6.97$ μm ($n = 5$) and the spatula at the apical end of the setae a length of $3.36 \pm 0.32$ μm ($n = 5$) and a width $1.62 \pm 0.13$ μm ($n = 5$). The flattened terminal spatulae are corrugated dorsally, but flat ventrally (Figure 4F).

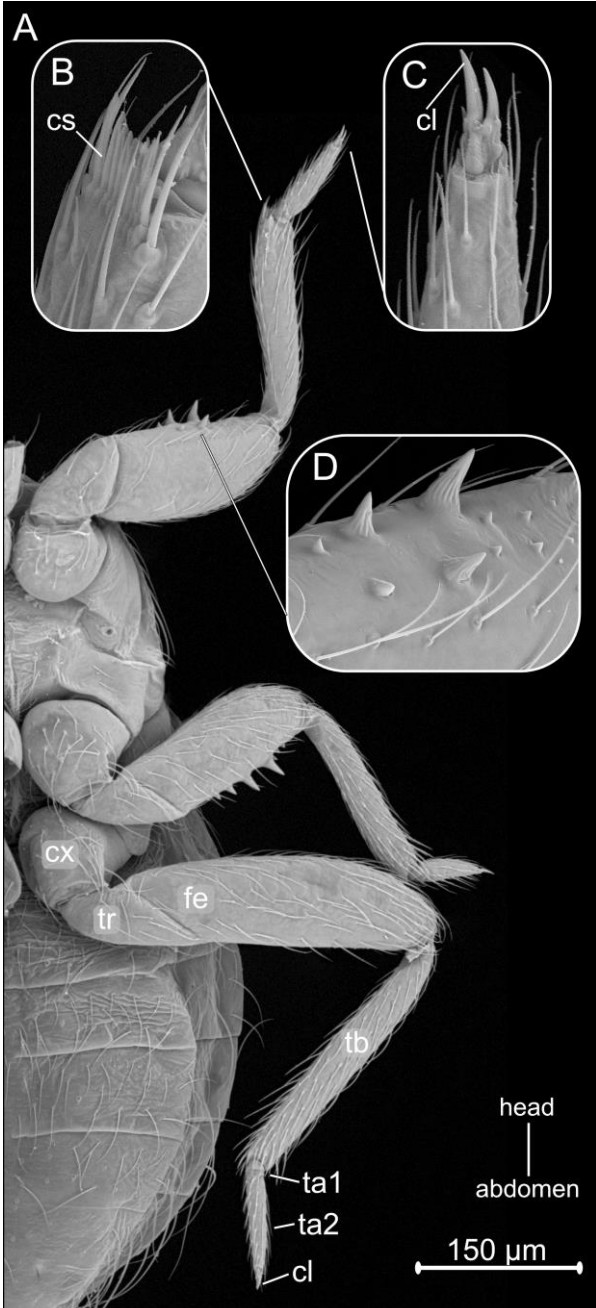

**Figure 3.** SEM micrograph of *Embiophila myersi*, nymph. (**A**) Thorax, legs and abdomen ventral view. (**B**) 'Grooming' setae, apical at the prothoracic tibia. (**C**) Prothoracic claws. (**D**) Rows of spines (raptorial) on the prothoracic femur. Abbreviation: cl—claw; cs—cleaning structure; cx—coxa; fe—femur; ta—tarsomere; tb—tibia; tr—trochanter.

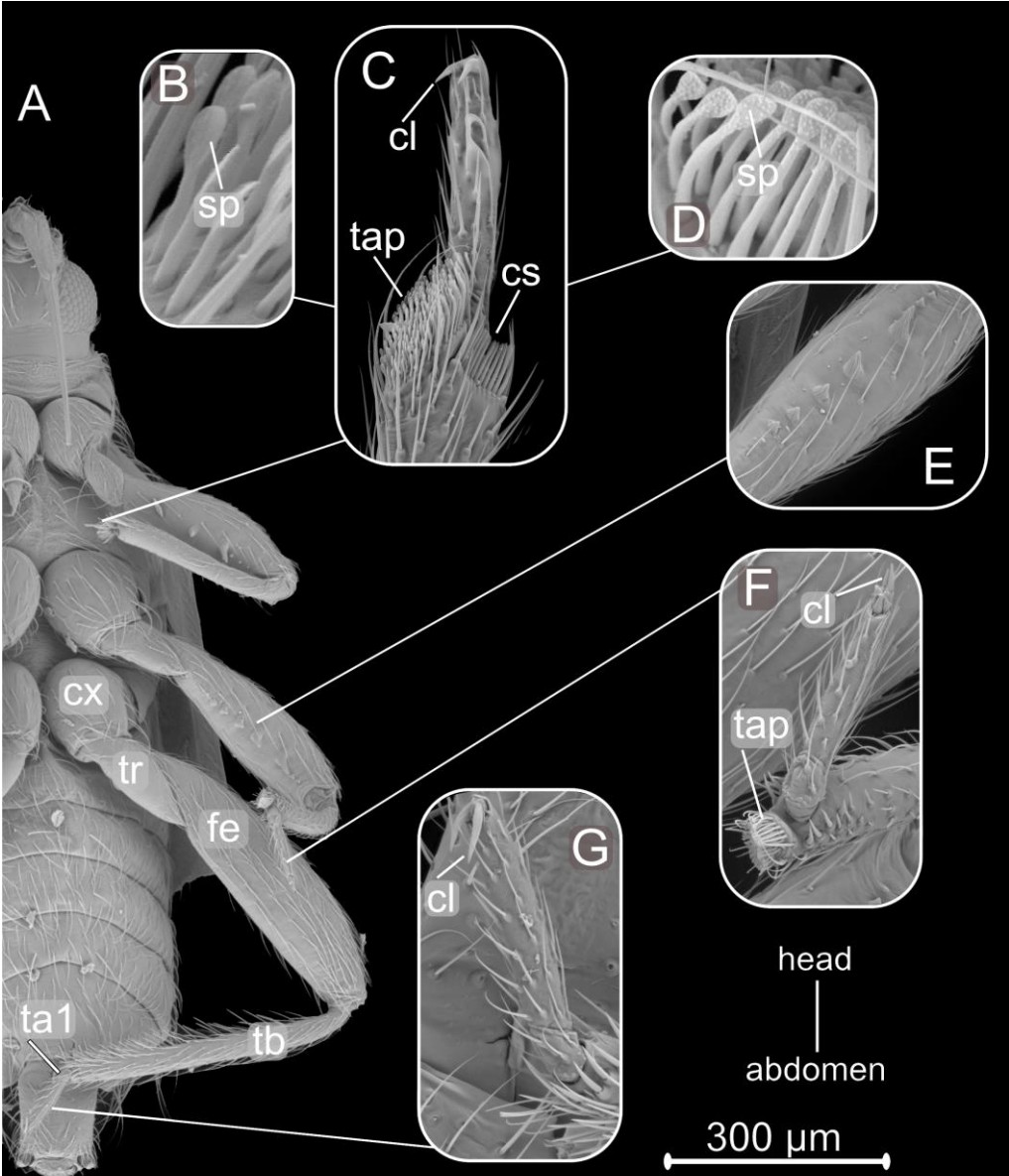

**Figure 4.** SEM micrograph of *Embiophila myersi*, adult. (**A**) Head, thorax, legs and abdomen ventral view. (**B**) Ventral view of the spatula of the tibial adhesive pad. (**C**) Apical part of the prothoracic leg showing claws, cleaning structure and the tibial adhesive pad. (**D**) Dorsal view of the spatulae with corrugated surface. (**E**) Row of spines (raptorial) on the mesothoracic femur. (**F**) Apical part of the mesothoracic leg showing claws and tibial adhesive pad. (**G**) Apical part of the metathoracic leg showing the claws. Abbreviation: cl—claw; cx—coxa; cs—cleaning structure; fe—femur; sp—spatula: ta—tarsomere; tap—tibial adhesive pad; tb—tibia; tr—trochanter.

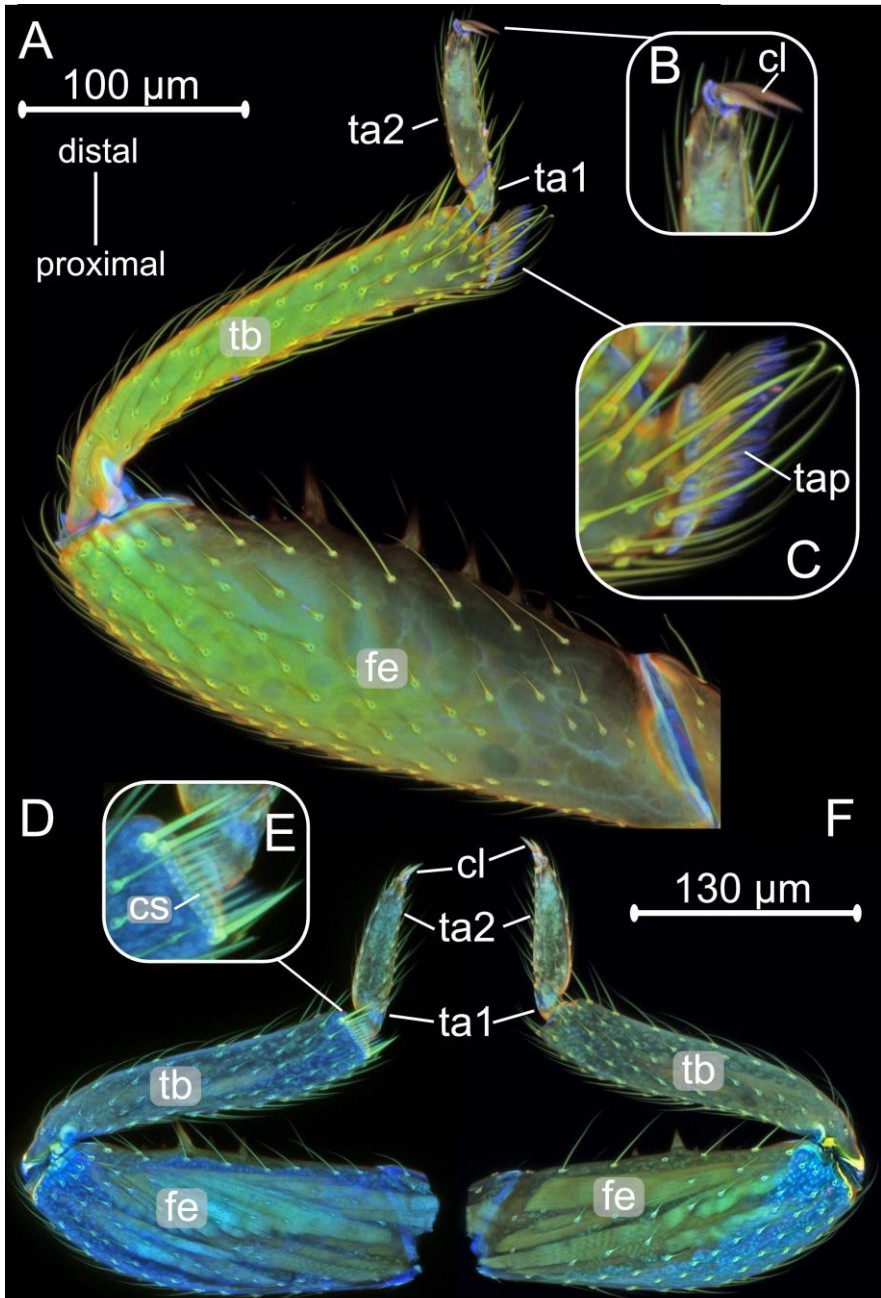

**Figure 5.** CLSM maximum intensity projections of *Embiophila myersi*. (**A–C**) Adult (**D–F**) nymph. (**A–C**) Prothoracic leg, anterodorsal view. (**B**) Claws. (**C**) Tibial adhesive pad. (**D,E**) Prothoracic leg, anterior view. (**E**) 'Grooming' setae (**F**) Prothoracic leg, posterior view. Abbreviation: cl—claw; cs—cleaning structure; fe—femur; ta—tarsomere; tap—tibial adhesive pad; tb—tibia.

### 3.2. Attachment Performance of Adult Bugs

The attachment performance was investigated by measuring the critical tilting angles (cta) on substrates with varying surface roughness. The substrate roughness had a significant influence on the attachment of the bugs to the substrates (Figure 6, One-way ANOVA, d.f. = 5, F = 41.729, $p < 0.001$, N = 7). The strongest attachment was found on the substrates with 1 μm and 35 μm roughness, both revealed significantly higher tilting angles than the other substrates (Holm–Šídák–Posthoc test, each combination $p < 0.001$), but not different from each other ($p = 0.927$). The cta were 70.57 ($\pm$16.87)° (mean $\pm$ s.d.) on 1 μm and 71.14 ($\pm$14.58)° on 35 μm roughness. Medium critical tilting angles were found for the other structured substrates. The bugs started sliding at 30.43 ($\pm$10.11)° on 0.3 μm

roughness, at 27.57 ($\pm$11.73)° at 3µm roughness and at 23.57 ($\pm$7.93)° at 12 µm roughness. There was no statistically significant difference between the attachment performance on these three substrates (Holm–Šídák–Posthoc test, 0.3 µm vs. 3 µm: $p = 0.875$; 0.1 µm vs. 12 µm: $p = 0.722$; 12 µm vs. 3 µm: $p = 0.890$), but they were different from the attachment performance of 1 µm and 35 µm, as well as glass (Hol—Šídák–Posthoc test, glass vs. 12 µm: $p = 0.003$; all other combinations $p < 0.001$). The lowest attachment was present on smooth glass, i.e., no adhesion at all.

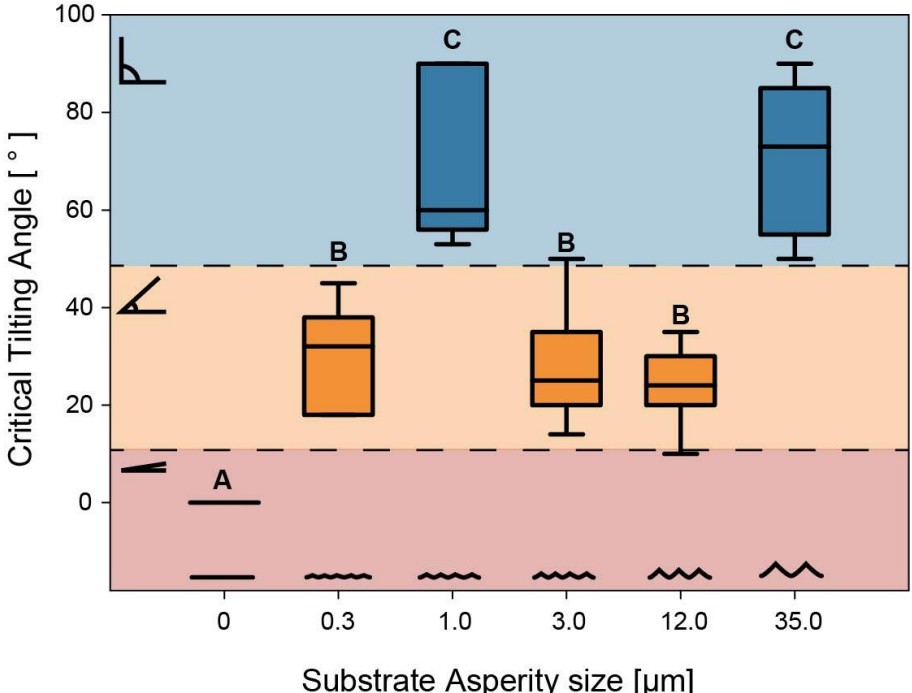

**Figure 6.** Attachment performance. Boxplots showing the critical tilting angles, at which the individuals started to slide during the substrate tilting experiments. Letters indicate statistical differences. Different letters are significantly different from each other (ANOVA on ranks, Holm-Sidak post hoc test $p < 0.05$).

*3.3. Choice Tests: Trials with Fibers Offered*

Typically, the bugs stayed closer to the walls of the arena when they were in a zone that was empty than they chose to when silk was present (Wilcoxon rank sum test: Z = 4.87, $p < 0.0001$; Figure 7). The bugs walked around without appearing to specifically orient to the silk or cotton fibers, as evidenced by the amount of time the bugs clung to the outer walls, even when fibers were nearby. A few bugs did not find the fibrous substrate until the end of the trial, contributing to the high variability in time spent in zones containing such materials (Figure 8; Supplementary Figure S1). When presented with a choice of cotton or silk in the arena, the time spent did not differ significantly between cotton and actual silk (Kruskall–Wallis ANOVA $\chi^2 = 6.59$, $p = 0.037$; Wilcoxon post hoc test: Z = 1.679, $p = 0.09$). This was also true for clean silk and silk with fecal pellets presented as a choice to the bugs (Wilcoxon post hoc test: Z = 0.113, $p = 0.91$). Once a bug wandered to silk or cotton, it tended to stay there, as reflected in the significantly greater amount of time spent in those zones compared to the empty zones overall (Kruskall–Wallis ANOVA $\chi^2 = 10.96$, $p = 0.0009$; Wilcoxon post hoc test: Z = 5.11, $p < 0.0001$; Figure 8). Heat maps also demonstrate the wandering, edge-seeking and tendencies to stay in the silk or cotton once discovered (Supplementary Figures S1 and S2).

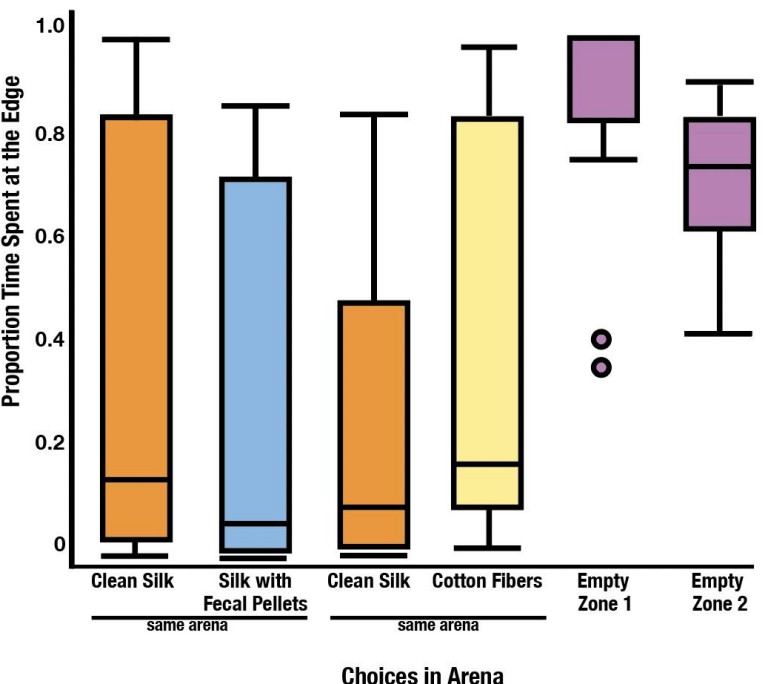

**Figure 7.** Box-and-whisker plots showing proportion of the time that individual adult E. myersi spent along the edges of arena walls. Clean silk and silk with fecal pellets were presented in different zones with two empty zones in the arena also as choices for each experiment (*n* = 10). Clean silk and cotton wool fibers were presented at the same time also with two empty zones for a second experiment (*n* = 9). Data from the empty zones that were presented along with the different substrates are combined, as the times along the walls in empty zones did not differ for these different experiments. Outliers are shown as purple circles.

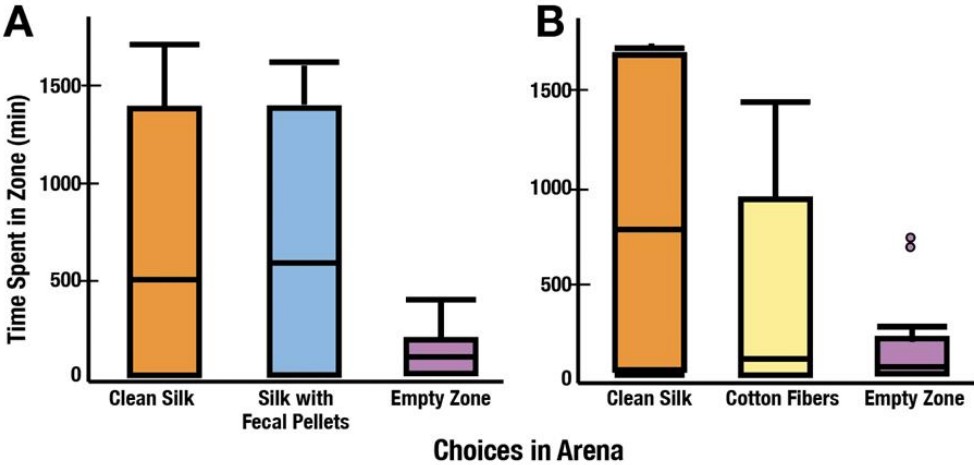

**Figure 8.** Amount of time, shown as box-and-whisker plots, that individual adult E. myersi spent in zones with different choices. (**A**) Zones included clean silk, silk with fecal pellets and two empty zones (*n* = 10). (**B**) Zones contained cotton wool fibers, clean silk and two empty zones (*n* = 9). Outliers are purple circles.

*3.4. Choice Tests: Wandering*

　　Wandering was at a higher level if the bugs moved into an empty quadrant because, alternatively, when a substrate was present, they usually moved onto or underneath to settle and stay (Figure 9). Bugs presented with cotton fibers and clean silk as alternative choices showed a higher rate of transitioning into empty zones but once they located a zone with one of the substrates they tended to stay (reflected in the lower value shown in red

in Figure 9). Transitions out of the empty zones were not significantly different for trials, when lichens or a fiber substrate were offered in the other quadrants (Wilcoxon rank sum test: Z = 1.085, *p* = 0.279). The number of transitions into zones with the substrates were higher when lichens were present than when a fiber (cotton or silk) was present (Wilcoxon rank sum test: Z = 2.02, *p* = 0.04; Figure 9). This result reflected the tendency for the bugs to wander away again when lichens were encountered only to come back later compared to when silk was discovered. When they wandered into a zone with silk or cotton, they tended to stay there, and the number of transitions into such zones was lower than for zones stocked with lichens. Detailed records of transitions are shown as red and blue arrows in Supplementary Figure S3.

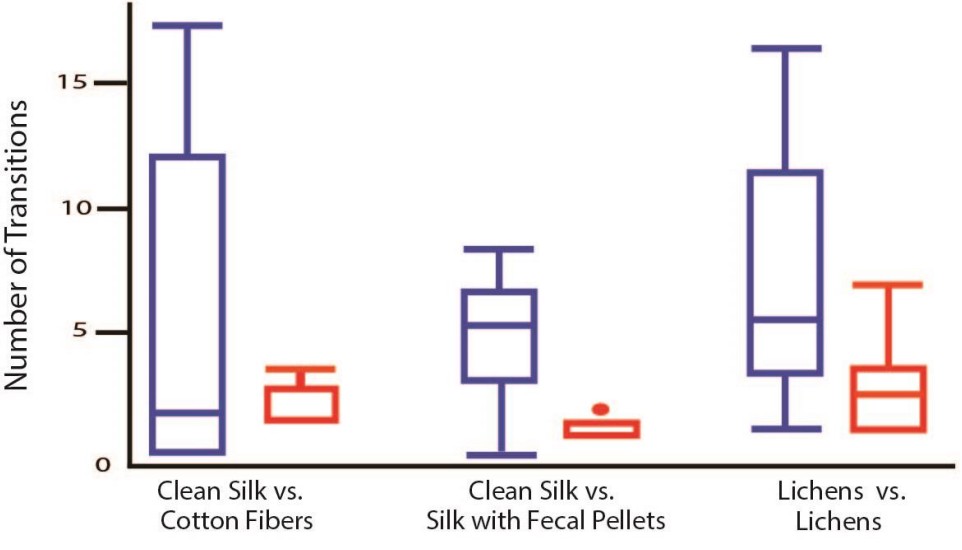

**Figure 9.** Transitions of E. myersi individuals between the presented four zones (see Figure 2) Blue represents transitions, where a bug walks out of empty zones; red represents transitions when a bug walks out of a zone with any of the substrates presented. See Supplementary Figure S3 for a summary of each trial.

## 4. Discussion

### 4.1. Behavioral Choice for Silk

In all behavioral experiments, the bugs actively wandered in the arena, but they tended to settle once they encountered a fibrous substrate. This tendency is detectable in the heat maps (Figure 2). The most frequented spots are associated with substrates and are not in the open areas. Lichens were not as much frequented even though they would have provided a natural hiding place. The bugs often walked into and under the lichen pieces but tended to emerge again to wander. These results suggest that the bugs are actively seeking silk as habitat, as expected, given the recorded association with host silk in the field [53]. Even in the short term, as in these trials, hiding did not appear to be the general motivation for site selection. Their ability to walk easily around the arena also indicates that they do not require a fibrous substrate in order to travel. This corroborates our observations of their ability to walk on substrates of various roughnesses. Surveys have not been conducted in Trinidad in the field to know how often they are wandering outside the silken embiopteran domiciles. Dispersal out of silk could be part of their life cycle, however, as they are dependent of embiopteran colonies, which can be temporary and might require them to migrate into another colony after collapse or abandonment of the previous gallery.

Although *Embiophila* spend their whole lifetime in the galleries of webspinners, our results do not suggest that they are particularly dependent on the silk itself, but prefer

any kind of fibrillar substrate, as evidenced by their willingness to settle in cotton fibers. Nevertheless, these bugs explicitly look for such kinds of substrates. The strong association with webspinners is rather driven by exploitation of the galleries for protection than using webspinners as a food source [35]. The strong spines on the ventral surface of the femur of both the pro- and mesothoracic legs of nymphs and adults suggest a predatory lifestyle, as already pointed out by Carayon [35] and Ross [8]. However, their actual food preferences still remain in question. It seems that the bugs are rather opportunistic and do not solely depend on embiopterans, if at all [8,35]. It is likely that *Embiophila* primarily feed on other small invertebrates that intrude into the galleries of the webspinners, such as mites, and, hence, seek shelter in the silk galleries, where they find not only protection, but also a source for food.

### 4.2. Morphological Attachment Constraints

While an arolium, a lobed pretarsal attachment pad, is present in the ground plan of most polyneopteran insects, and even Coleorrhyncha have one, this attachment pad has been reduced in true bugs (Heteroptera; [61]). Its absence is considered a synapomorphy of Cimicomorpha + Pentatomorpha [62], as it is reduced to a peg-like structure in Cimicomorpha [63]. Instead, different other attachment devices evolved within Heteroptera, including pulvilli and parempodia [64–66]. *Embiophila myersi* possesses a strongly reduced tarsal attachment system, as in most other cimicomorphan bugs of which the tarsal morphology is known in detail [67,68]. No tarsal attachment devices are present, except for paired pretarsal claws and a few tenent setae on the tarsi which can potentially generate attachment as well (shown for a pentatomid bug: [65]. Although parempodia and similar tarsal setae are reported to generate attachment as well [65], the contribution of the two small parempodia to the overall attachment is probably quite low in *Embiophila*. Instead of the tarsus, the tibia of many adult Cimicomorpha bears a tibial attachment pad (TAP) consisting of many spatulate tenent setae [67,68]. This pad is called fossula spongiosa in Reduvioidea [69–71], and is absent in juvenile individuals of the same species [67]. The fossula spongiosa is present in other true bugs as well [43,72,73], and different authors showed that a similar structure evolved at least three times independently in Heteroptera [24,74–77]. As the fossula spongiosa of Reduvioidea and the equivalent attachment pad in Plokiophilidae are not homologous, the plokiophilid structure will be called tibial adhesive pad (TAP) herein. However, the functional principle of both structures is the same: deformable setae enable adjustment of the structures to the surface profile and the spatulae at the tips make close contact to the substrate to generate adhesion [46]. The TAP likely provides the greatest contribution to attachment in *Embiophila* on smooth surfaces, where claws can not engage (see *Attachment Performance*), whereas the claws are the only other attachment device with major importance for this insect.

The asymmetric claws are conserved in Plokiophilidae [34]; however, their functional role for locomotion on silk remains to be experimentally tested. The outwards-facing (lateral) claw is smaller. This could be beneficial for adapting to the cylindric nature of the silk domiciles of webspinners. The apomorphic presence of asymmetry of the claws for Plokiophilidae could indicate a common adaption to the silk domiciles of the hosts [34,36]; however, all plokiophilids bear asymmetric claws, but only *Embiophila* settles in the nests of webspinners. Nevertheless, it is possible that the claw asymmetry evolved due to the specification towards silk domiciles in general.

### 4.3. Attachment Performance

The asymmetric claws of *E. myersi* are well suited for mechanical interlocking with their preferred substrate silk, or similarly frequented fibrous substrates, as discussed above. On non-fibrous substrates with low roughness, however, the main contributing devices are the TAPs. The material composition of the TAPs cuticle is very characteristic for hairy attachment systems; the material gradient of the setae—soft, likely resilin-dominated apical tips and stiff, more sclerotized bases—is well known for attachment systems of insects [52,65].

As described by Peisker and colleagues [78], this gradient within the setae allows for an optimized contact formation with the substrate to generate attachment forces. Here, the combination of soft tips that adapt to varying roughnesses on the substrate and stiff bases to generate mechanical stability facilitates efficient attachment [78,79]. Hairy adhesive devices similar to the TAP of *E. myersi* usually perform good on smooth substrates [46,80]; however, the plokiophilids were not able to adhere to smooth glass at all. The performance of the TAP on smooth substrates is ambiguously reported in other Cimicomorpha as well. Bed bugs (*Cimex lectularius* Linnaeus, 1758) do not well adhere to plastic and glass surfaces according to Hottel et al. [81], while Reinhardt et al. [68] measured significant attachment forces on different smooth substrates (smooth epoxide resin, hydrophilic glass and silanized, hydrophobic glass). Nevertheless, these forces also differ between the sexes of *C. lectularius* [68]. While males adhered better on hydrophilic substrates, females performed better on hydrophobic ones. This is hypothesized in the literature to be a result of the different substrates the animals attach to during mating [67,68,71,82]. The position of the TAP on the distal tip of the tibia appears to be beneficial for grasping non-planar substrates. At least it is suboptimal for attaching to flat substrates, as indicated by the tilting tests. On curved substrates, the position of the TAP might be more suitable for engagement with the substrate.

Substrate geometry might play a role for contact formation of the tarsal segments [83], which would also agree with the hypothesis that the TAPs are of particular interest for the attachment during mating. The slight convex curvatures typically found on the female body might be required for proper contact formation of the TAP, thus explaining the absence of adhesion on plane glass found here. As a supplemental test, adult plokiophilids were placed in a glass tube with 5.0 mm diameter. Within these tubes, they still attached at angles of 90° tilting and even resisted tapping of the tube (pers. obs. JER). This observation indicates that a concave surface curvature is beneficial for attachment of these insects compared to plane surfaces. Given that the silk domiciles are at least in large parts tubelike concave, this is a possible adaptation of the true bugs for the structure of the galleries. In Zoraptera, specialized tarsal joints aid in torsion of the tarsus to enable movement along small crevices [84] and can potentially enable contact formation of the tarsi with the inner perimeter of tubelike constructions. However, we did not find indications for a similar adaptation in the tarsi of *E. myersi* (Figure 4), except for a rather similar shape of the tarsomeres between Zoraptera and *E. myersi* and a comparably mobile base of the tarsus (Figure 4D). Kim et al. [67] additionally found differences between the adhesive capability of *C. lectularius* and another closely related *Cimex* sp. with similar morphology of the TAP interfering with a comparison of the attachment performance of *E. myersi* with bed bugs in general. The micromorphology of *E. myersi* and *Cimex* spp., however, is very similar and thus does not explain the functional differences between the taxa. The spatulae of *E. myersi* are corrugated dorsally and flat ventrally (Figure 4), as described for *Cimex* spp. as well [68], and their size is similar as well.

Most insects for which the attachment performance was tested in experimental studies showed the lowest force in the range of 0.05 μm and 3 μm roughness [64,85–88]. In general, for both fibrillary and smooth attachment systems, the attachment performance was the highest on smooth substrates due to the function of the attachment pads and on rough substrates because of the claws [89]. The attachment performance of *E. myersi* differs from these results obtained from other insects. First of all, the best attachment performances were found on 1 μm roughness and 35 μm roughness (Figure 6). Claws of insects can sufficiently engage with the substrate if the diameter of the surface asperities is smaller than the diameter of the tip of the claws [90]. The finer claw tips of *E. myersi* (~670 nm diameter) compared to other insects' claws used in similar experiments could therefore explain the ability to generate attachment on micro-rough substrates. Fine claw tips are useful for interlocking with the silk in their natural environments. The diameter of the silk of webspinners is rather small compared to silk of other invertebrates [40]. The diameter of the fiber bundles of *A. urichi*, the host webspinner of *E, myersi*, ranges around 800 nm [40,41].

It makes sense that the tips of the claws of. *E. myersi* are smaller to interlock with the gaps of the silk fibers [87]. A straight claw, as observed in the species examined here, can also be beneficial for the detachment process from the silk, as the retraction of the claw can easily disengage the claw from the substrate.

In addition to the morphology of the claw, structural features of the TAP can contribute to the attachment on micro-rough substrates similar to the spatulate tarsal setae of other insects [46]. We hypothesize that there was no attachment on smooth flat glass at all, because of the geometry of the substrate. However, two studies on bed bugs with a very similar TAP [67,68], show that these insects generated attachment on glass and even provide photographs of the TAP of *Cimex hemipterus* (Fabricius, 1803) in contact with a smooth petri dish [67]. One reason for this discrepancy could be the different tarsal morphology between *Cimex* spp. and *E. myersi*. The combination of shorter, less curved tarsi and claws could be less favorable to make contact with a planar substrate. The corrugations on very large surface roughness, as on the 34 µm substrate, result in a more complex surface topography for the dimension of the small tarsi of *E. myersi*. Although macroscopically planar, the surface profile at 34 µm roughness is constituted of both concave and convex structures that can also enable the TAP to be brought into contact.

The dimension of the single spatulae of the TAP is another point that can potentially explain the attachment performance on the 1 µm rough substrate. The terminal spatulae of the TAP in *E. myersi* is smaller (length 3.36 ± 0.32 µm, width 1.62 ± 0.13 µm) than for most species of which data on the spatula tip were available and that showed the minimum attachment performance in this range of surface roughness. For example, the spatulae of the chrysomelid beetle *Gastrophysa viridula* (De Geer, 1775) were reported to have a width of ~4 µm [91] and those of the chrysomelid *Leptinotarsa decemlineata* (Say, 1824) were reported to have a width of ~7 µm (measured from Figure 2 in [85]). Both species represent insects with fibrillary tarsal attachment structures and a performance minimum at low surface roughness [85,89]. Smaller spatulae are likely an adaptation towards surface roughness, with a smaller nominal asperity size of importance for the bugs. Indeed, the fiber bundles of silk of the known webspinner host are below 1 µm in diameter [40,41]. An adaptation to this range could explain the comparably better attachment performance of *E.myersi* on the substrate with 1 µm roughness. As the small spatulae are almost in the range of the substrate asperities, they are able to make sufficient actual contact with the surface [91].

Nymphs of *E. myersi* do not possess TAPs at all and, consequently, their attachment ability is restricted to interlocking of the claws with the fibrous substrates. It is probable that they benefit from the shelter of the silk galleries, and primarily use their claws for engagement with the silk. The presence of the TAP being limited to adults might be due to two reasons: either the TAP is involved in mating, or only adults serve as dispersal states and might be urged to leave the secure galleries of the webspinners. A role of the TAP for dispersal appears unlikely, as the observations on the functional morphology of the attachment system is well adapted to living in the galleries and shares less similarities with the adhesive systems of free living insects (e.g., the size of the spatulae). To further investigate the role of the TAP for reproduction, more detailed research is required to investigate sexual dimorphism in the functionality of the TAP, as reported for *Cimex* spp. [68].

## 5. Conclusions

The silk-loving bug *E. myersi* is strictly associated with silk galleries of webspinners. However, our choice trials indicate that they do not actively search for the presence of webspinners, but frequent any fibrous material. This finding supports the mutualistic nature of the embiopteran-embiophilan community found in the silk galleries. While the leg morphology suggests a raptorial lifestyle, the behavior of the true bugs apparently focuses on the shelter provided by the webspinner galleries. The ignoring of webspinner traces supports earlier theories that *Embiophila* feed on mites and other intruding arthropods.

The domiciles of webspinners are dominated by silk, but occasionally, the bug might be urged to locomote on other substrates as well, and during mating, males need to attach

to the surface of females. While the sharp claws interlock well with fibrous substrates such as the silk, the tibial attachment system of *E. myersi* is quite specialized towards substrates with certain surface roughness and maybe substrate curvature, but surprisingly, not specifically for attachment on silk.

**Supplementary Materials:** The following supporting information can be downloaded at: https://www.mdpi.com/article/10.3390/d15030415/s1, Figure S1: Heatmaps summary 1; Figure S2: Heatmaps summary 2; Figure S3: Transition graph; Table S1: Test for bias; Table S2: Summary of Behavior Trials.

**Author Contributions:** Conceptualization, J.S.E., T.H.B. and S.B.; methodology, J.S.E., S.B., S.N.G. and T.H.B.; software, J.S.E. and T.H.B.; validation, J.S.E., S.N.G. and T.H.B.; formal analysis, J.R.H., N.S. and T.H.B.; investigation, J.R.H., N.S., S.B. and T.H.B.; resources, J.S.E. and S.N.G.; data curation, J.S.E., S.B. and T.H.B.; writing—original draft, T.H.B., J.S.E. and S.B.; writing—review and editing, J.R.H., N.S. and S.N.G.; visualization, J.S.E., S.B. and T.H.B.; supervision, J.S.E. and S.N.G.; project administration, T.H.B.; funding acquisition, J.S.E. and S.B. All authors have read and agreed to the published version of the manuscript.

**Funding:** S.B. was directly supported by the DFG grant BU3169/1–2. Acquisition of the Noldus Ethovision System was funded by a Collaborative Teaching and Technology Grant from Santa Clara University awarded to J.S.E. and colleagues in the Departments of Biology and Neuroscience.

**Institutional Review Board Statement:** Not applicable.

**Informed Consent Statement:** Not applicable.

**Data Availability Statement:** All data supporting our findings are presented in the paper and the Supplementary Materials, respectively.

**Acknowledgments:** T.H.B., S.N.G. and S.B. thank the Functional Morphology & Biomechanics working group (Kiel University, Germany), especially Benedikt Josten for his help in the analysis of the data and Esther Appel for technical support during microscopy. The Maker Lab of SCU's School of Engineering provided access for J.R.H. to the 3D Printer for producing the test arenas.

**Conflicts of Interest:** The authors declare no conflict of interest.

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
