# Peer review of "Morphological and Behavioral Adaptations of Silk-Lovers (Plokiophilidae: Embiophila) for Their Lifestyle in the Silk Domiciles of Webspinners (Embioptera)"

_diversity, doi:10.3390/d15030415_

Round 1

Reviewer 1 Report

The present manuscript (MS) combines morphological, performance and behaviour experiments, which combined allow conclusions about the life history of silk-lovers. This combination of approaches is highly needed to disentangle the phenotypic diversity of organisms. The Experiments are well done and clearly presented.

I only have a few points I would like to raise to the attention of the authors, which hopefully will further improve the MS.

In my opinion, the last sentence of the Abstract could be removed, as it does not really contribute to the summary of the study. If the authors wish to keep it, the second halve of the sentence should be reformulated. I agree that the laboratory nature of the study is highly beneficial in this context, and that field studies for these small animals would be highly challenging. But claiming that field studies would not be possible at all is, in my opinion, a statement that is far too bold – and also not necessary to highlight the merits of this study.

The structure of the introduction could be improved and made more consistent throughout: 1) in the first paragraph the authors first describe permanent shelters of colony building insects in general (till LL36), then name webspiders as one peculiar case (LL36-39). Next, they describe how other species use these homes (LL41-43), but it is at this stage not clear, if this refers to permanent shelters in general, or those of the webspiders in particular. Then the authors explicitly summarize the knowledge of other species associated with the webs of webspider (LL43-50). I think it would be clearer and more consistent to first describe and summarize the phenomenon of permanent shelter including other species using these and then present webspiders as a special case and summarize what is known about the species associated with these. 2: At the end of the first paraphraph the authors state the silk is ‘in general a challenging substrate for locomotion’ (LL67ff), but then the next paragraph (LL76-85) constitutes a general introduction to the importance of locomotor substrates for terrestrial animals. I strongly suggest to re-structure these two parts into one paragraph starting with the importance of substrates for terrestrial locomotion followed by the information presented on silk as substrate. Also, in the paragraph LL76-84 you switch between talking about animals in general and insects in particular without a clear structure, which makes it a bit difficult to follow. And, if you generalize this part to animals in general – which I agree is a good thing to do here – you should cite a few more references on locomotion and substrate interactions in non-insect systems.

 In the results you report a measurement of tip sharpness, but no information on how this was measured and on how many individuals. This need to be stated in the methods. And please provide exact p-values in results section for the attachment force, not just lager or smaller than 0.05, as this helps the reader to assess the validity of the conclusions. And, I would suggest presenting plots equivalent to those presented in fig. 7 and 8 also for the data of the third experiment with lichen as a substrate in the supplement, as I think it could be interesting for the reader to compare those to the plots in fig. 7 and 8.

The hypothesis presented on why TAP are present only in adults are, as ideas, convincing and quite intriguing. Albeit the second hypothesis that this is associated with dispersal, contradicts the suggested interpretation of smaller spatulae as adaptations to the small diameter of the webspider silk mentioned just before. This contradiction should not be left uncommented. And the first hypothesis, their putative importance for mating would imply a sexual dimorphism in this trait as reported for bed bugs in the literature. Although is understandably beyond the scope of this study to test for such dimorphism in either morphology or performance, the authors should, in my opinion, discuss this more detail, maybe as an outlook for future experiments.

Detailed comments:

LL35: I would suggest naming the higher order clades of the examples provided with references here. Otherwise, the sentence remains somewhat obscure to the reader.

LL47: I don’t think that commitment is a well fitting word here. Better use association or something similar.

LL53/54: As the study does not deal with bedbugs, the sentence is unnecessary here and should be removed.

LL72: please elaborate on how exactly the silk of Embioptera differ from that of spiders.

LL68-74: The argument, as it is presented here, is contradicting and inconsitent. You first state that ‘silk in general’ is a challenging substrate for locomotion’ and to support this argument you elaborate on the risk of being glued to spider silk. Then you state that the silk of webspiders is ‘not intended to capture insects and, hence comes with different properties than spider silk’, rendering your previous argument not applicable to webspider silk. In the present state of the MS the first part on spider silk could be removed completely, while there is no explanation of justification why webspider silk is a challenging substrate.

LL138: Why was the temperature kept constant at 27C? Would not a variation in temperature between daylight and nighttime be more natural for the animals?

LL189: Time spend where or on what? The sentence is incomplete.

Figure 4: I would suggest to switch panels B and C as B is a detail of C. Also, you refer to cleaning structures in the figure, but no explanation or justification is given in the text, why the authors interpret these structures as such. This needs to be provided, including a reference for the interpretation.

LL290: G not E

Figure 6: The panels B-D are redundant, as these structures are already shown in fig. 4, and these panels are also not referenced anywhere in the text. They should, thus, be removed.

LL328: did not differ significantly between or among what? Please be precise when reporting results.

Figure 9: It is irritating that the box plots in this figure have a different design than the boxplots in fig. 7 & 8. It would look better if you would keep the figures in one consistent design. Also, there is a typo: change ‘Clean SIlk' to ‘Clean Silk’.

LL371: Where transition when bugs walked out of a zone with any substrate removed from this dataset (if yes, why?), or did they never occurred?

LL390-392: The logic of the argument here is unclear to me.

L433 & 438: Please provide a reference for the conserved nature of asymmetric claws in plokiophilidae.

LL442-445: This sentence also lacks a reference.

LL488/489: This sentence is redundant with LL 493/494, and also does not fit well here. Please remove.

LL501/502 Reference missing.

LL530-533: The statement is redundant with LL502/503. Please remove.

Author Response

Thank you for your feedback. Below is a point by point response to all comments and how we followed the suggestions.

The present manuscript (MS) combines morphological, performance and behaviour experiments, which combined allow conclusions about the life history of silk-lovers. This combination of approaches is highly needed to disentangle the phenotypic diversity of organisms. The Experiments are well done and clearly presented.

I only have a few points I would like to raise to the attention of the authors, which hopefully will further improve the MS.

Answer: Thank you!

In my opinion, the last sentence of the Abstract could be removed, as it does not really contribute to the summary of the study. If the authors wish to keep it, the second halve of the sentence should be reformulated. I agree that the laboratory nature of the study is highly beneficial in this context, and that field studies for these small animals would be highly challenging. But claiming that field studies would not be possible at all is, in my opinion, a statement that is far too bold – and also not necessary to highlight the merits of this study.

Answer: Agreed, we changed the second half sentence.

The structure of the introduction could be improved and made more consistent throughout: 1) in the first paragraph the authors first describe permanent shelters of colony building insects in general (till LL36), then name webspiders as one peculiar case (LL36-39). Next, they describe how other species use these homes (LL41-43), but it is at this stage not clear, if this refers to permanent shelters in general, or those of the webspiders in particular. Then the authors explicitly summarize the knowledge of other species associated with the webs of webspider (LL43-50). I think it would be clearer and more consistent to first describe and summarize the phenomenon of permanent shelter including other species using these and then present webspiders as a special case and summarize what is known about the species associated with these.

2: At the end of the first paraphraph the authors state the silk is ‘in general a challenging substrate for locomotion’ (LL67ff), but then the next paragraph (LL76-85) constitutes a general introduction to the importance of locomotor substrates for terrestrial animals. I strongly suggest to re-structure these two parts into one paragraph starting with the importance of substrates for terrestrial locomotion followed by the information presented on silk as substrate. Also, in the paragraph LL76-84 you switch between talking about animals in general and insects in particular without a clear structure, which makes it a bit difficult to follow. And, if you generalize this part to animals in general – which I agree is a good thing to do here – you should cite a few more references on locomotion and substrate interactions in non-insect systems.

Answer: We reshaped the introduction by adding a clearer outline of why the sequence of information is of concern and put more focus on the silk construction of webspinners, as these are the only actual relevant silk constructions for this study. Therefore we reduced the focus on spider silk as a substrates, while stronger emphasizing general the substrate attachment system relationship.

In the results you report a measurement of tip sharpness, but no information on how this was measured and on how many individuals. This need to be stated in the methods.

Answer: The information has been specified.

And please provide exact p-values in results section for the attachment force, not just lager or smaller than 0.05, as this helps the reader to assess the validity of the conclusions.

Answer: The p-values have been specified.

And, I would suggest presenting plots equivalent to those presented in fig. 7 and 8 also for the data of the third experiment with lichen as a substrate in the supplement, as I think it could be interesting for the reader to compare those to the plots in fig. 7 and 8.

Answer: Thank you for the suggestion. Unfortunately the analysis of the videos using Ethovision is very tie consuming, as the small size of the bugs requires a lot of manual tracking. Consequently, the time granted for revision by the journal does not allow for it. Although these results can potentially be interesting, we did not observe major patterns that would add to the information already presented.

The hypothesis presented on why TAP are present only in adults are, as ideas, convincing and quite intriguing. Albeit the second hypothesis that this is associated with dispersal, contradicts the suggested interpretation of smaller spatulae as adaptations to the small diameter of the webspider silk mentioned just before. This contradiction should not be left uncommented. And the first hypothesis, their putative importance for mating would imply a sexual dimorphism in this trait as reported for bed bugs in the literature. Although is understandably beyond the scope of this study to test for such dimorphism in either morphology or performance, the authors should, in my opinion, discuss this more detail, maybe as an outlook for future experiments.

Answer: Thank you for pointing out. We added both aspects to the discussion.

Detailed comments:

LL35: I would suggest naming the higher order clades of the examples provided with references here. Otherwise, the sentence remains somewhat obscure to the reader.

Answer: Added.

LL47: I don’t think that commitment is a well fitting word here. Better use association or something similar.

Answer: Changed

LL53/54: As the study does not deal with bedbugs, the sentence is unnecessary here and should be removed.

Answer: Changed

LL72: please elaborate on how exactly the silk of Embioptera differ from that of spiders.

Answer: We specified the properties of embiopteran silk, but also removed the explicit notion of spider silk as it is not of major importance for the animals investigated here.

LL68-74: The argument, as it is presented here, is contradicting and inconsitent. You first state that ‘silk in general’ is a challenging substrate for locomotion’ and to support this argument you elaborate on the risk of being glued to spider silk. Then you state that the silk of webspiders is ‘not intended to capture insects and, hence comes with different properties than spider silk’, rendering your previous argument not applicable to webspider silk. In the present state of the MS the first part on spider silk could be removed completely, while there is no explanation of justification why webspider silk is a challenging substrate.

Answer: We removed the emphasis on spider silk, as indeed the only silk of concern is the special case of embiopteran silk.

LL138: Why was the temperature kept constant at 27C? Would not a variation in temperature between daylight and nighttime be more natural for the animals?

Answer: The webspinners and plokiophilids are from a region that does not experience a wide range of temperature variation throughout the year or even day to night. Webspinners can be found with eggs at any month of the year as well, reflecting the lack of variation. The webspinners are the primary cultures that JER is rearing and the bugs happened to be with them, so the conditions were meant to keep the webspinners healthy. The conditions chosen for the culture are based on decades of experience with breeding the webspinners and were continued as they have been working for the bugs as well during the last 30 years. The range for yearly average temperatures reported for the airport in Trinidad is 26.5 to 28.6°C. We kept a constant temperature of 27°C, which is similar to what one experiences there during any one day.

LL189: Time spend where or on what? The sentence is incomplete.

Answer: corrected: “For Trial 1 and for Trial 2, total time spent in the different zones of the arena was compared. Proportion of time spent along the edge in the different zones was similarly compared.”

Figure 4: I would suggest to switch panels B and C as B is a detail of C. Also, you refer to cleaning structures in the figure, but no explanation or justification is given in the text, why the authors interpret these structures as such. This needs to be provided, including a reference for the interpretation.

Answer: We do not agree. The insets B and D are opposite sides of the setae. We think this is more apparent in the present layout of the figure. We added information on the cleaning structures.

LL290: G not E

Answer: Corrected

Figure 6: The panels B-D are redundant, as these structures are already shown in fig. 4, and these panels are also not referenced anywhere in the text. They should, thus, be removed.

Answer: Changed

LL328: did not differ significantly between or among what? Please be precise when reporting results.

Answer: specified.

Figure 9: It is irritating that the box plots in this figure have a different design than the boxplots in fig. 7 & 8. It would look better if you would keep the figures in one consistent design. Also, there is a typo: change ‘Clean SIlk' to ‘Clean Silk’.

Answer: We adjusted the line thickness and design to make it more consistent. However, as the content of the figures is very different, the overall appearance of the figures also should differ to some extent.

LL371: Where transition when bugs walked out of a zone with any substrate removed from this dataset (if yes, why?), or did they never occurred?

Answer: They are included: red and blue boxes are transitions out of one type of zone. We have specified this now more clearly to avoid missunderstandings.

LL390-392: The logic of the argument here is unclear to me.

Answer: We now explain the argument with more detail.

L433 & 438: Please provide a reference for the conserved nature of asymmetric claws in plokiophilidae.

Answer: added.

LL442-445: This sentence also lacks a reference.

Answer: that’s a conclusion of the preceding part of the discussion. Specified now.

LL488/489: This sentence is redundant with LL 493/494, and also does not fit well here. Please remove.

Answer: removed

LL501/502 Reference missing.

Answer: added.

LL530-533: The statement is redundant with LL502/503. Please remove.

Answer: We do not completely agree, as the size of the silk fibres has in theory different effects for claws and/or spatulae. However, we reduced the details of the embiopteran silk fibres at this occasion to avoid repetition.

Reviewer 2 Report

The work submitted for review presents very interesting research conducted on live animals, which is a challenge. The obtained results are presented correctly and should be published.

Nevertheless, I believe that the presented manuscript needs some redrafting.

Some of the comments I proposed in the text (specifically on the first pages of the work), but in the case of the entire manuscript, the comments would have to be repeated.

Basic comments:

1. Captions under the photos should be much shorter, although they are part of the work, they should be captioned briefly, and the explanation of the presented figure should be in the main text, not under the figure itself.

2. lines 141-143: because there is a scale next to the image (Fig. 1A) (which should be more subtle), the information that the body length of an adult insect is about 2 mm is redundant.

3. The authors in some parts of the manuscript use mental abbreviations\colloquial phrases that should not be used in scientific works, example: 3D printed testing arena with the filter paper substrate showing a bug walking around - since this is a photograph, we cannot see the insect move or like Authors wrote walking around. Please, note the relative scale of walls and the bug - is it about the thickness of the walls (in this case irrelevant to the experiment), their height (which we can't see), or the size of individual cells? This sentence is redundant.

4. Lines 147-148: To determine, whether the bugs prefer to settle in silk or merely look for places to hide, different substrate choices were offered in 30 min trials - My suggestion: To determine the preferences of insects, indicating the need to find silk or the need to find a hiding place.....

I believe that some parts of the text (especially the figure captions) need redrafting.

It is not my intention to undermine the experiment or its results, but authors must understand that scientific work requires detailed and precise vocabulary - colloquially speaking: sometimes less is more.

Author Response

Thank you very much for your feedback. Below we list the point-by-point response to all comments:

The work submitted for review presents very interesting research conducted on live animals, which is a challenge. The obtained results are presented correctly and should be published.

Nevertheless, I believe that the presented manuscript needs some redrafting.

Some of the comments I proposed in the text (specifically on the first pages of the work), but in the case of the entire manuscript, the comments would have to be repeated.

Answer: Thank you for your suggestions, we have followed the suggestions below, the comments in the pdf file and tried to apply them to the whole manuscript.

Basic comments:

  1. Captions under the photos should be much shorter, although they are part of the work, they should be captioned briefly, and the explanation of the presented figure should be in the main text, not under the figure itself.

Answer: Agreed, we shortened the captions with repetitions and very long text.

  1. lines 141-143: because there is a scale next to the image (Fig. 1A) (which should be more subtle), the information that the body length of an adult insect is about 2 mm is redundant.

Answer: Agreed, omitted.

  1. The authors in some parts of the manuscript use mental abbreviations\colloquial phrases that should not be used in scientific works, example: 3D printed testing arena with the filter paper substrate showing a bug walking around - since this is a photograph, we cannot see the insect move or like Authors wrote walking around. Please, note the relative scale of walls and the bug - is it about the thickness of the walls (in this case irrelevant to the experiment), their height (which we can't see), or the size of individual cells? This sentence is redundant.

Answer: Agreed, we rephrased the figure legends, condensed the information and removed unnecessary details and redundant information.

  1. Lines 147-148: To determine, whether the bugs prefer to settle in silk or merely look for places to hide, different substrate choices were offered in 30 min trials - My suggestion: To determine the preferences of insects, indicating the need to find silk or the need to find a hiding place.....

Answer: Changed accordingly

I believe that some parts of the text (especially the figure captions) need redrafting.

Answer: We rephrased the figure legends and revised several parts of the text.

It is not my intention to undermine the experiment or its results, but authors must understand that scientific work requires detailed and precise vocabulary - colloquially speaking: sometimes less is more.

Answer: Thank you for the suggestions. We rephrased several part of the texts.

Comments in PDF file:

All comments and suggestions from the review PDF were followed as suggested.